# Distal Gastrectomy for Symptomatic Stage IV Gastric Cancer Contributes to Prognosis with Acceptable Safety Compared to Gastrojejunostomy

**DOI:** 10.3390/cancers14020388

**Published:** 2022-01-13

**Authors:** Nobuaki Fujikuni, Kazuaki Tanabe, Minoru Hattori, Yuji Yamamoto, Hirofumi Tazawa, Kazuhiro Toyota, Noriaki Tokumoto, Ryuichi Hotta, Senichiro Yanagawa, Yoshihiro Saeki, Yoichi Sugiyama, Masahiro Ikeda, Masayuki Shishida, Toshikatsu Fukuda, Keisuke Okano, Masahiro Nishihara, Hideki Ohdan

**Affiliations:** 1Department of Surgery, JA Onomichi General Hospital, Onomichi 7228508, Japan; fujikuni2292@gmail.com (N.F.); qqry3v7d@silk.ocn.ne.jp (S.Y.); 2Department of Perioperative and Critical Care Management, Graduate School of Biomedical and Health Sciences, Hiroshima University, Hiroshima 7398511, Japan; 3Center for Medical Education Institute of Biomedical & Health Sciences, Hiroshima University, Hiroshima 7398511, Japan; m-hattori@hiroshima-u.ac.jp; 4Department of Gastroenterological Surgery, Hiroshima Prefectural Hospital, Hiroshima 7340004, Japan; yuji_yamamoto1020@yahoo.co.jp; 5Department of Surgery, Kure Medical Center/Chugoku Cancer Center, Kure 7370023, Japan; thiroes@gmail.com; 6Department of Surgery, Hiroshima Memorial Hospital, Hiroshima 7300802, Japan; toyota@kkrhiroshimakinen-hp.org; 7Department of Gastroenterological Surgery, Hiroshima City Asa Citizens Hospital, Hiroshima 7308518, Japan; gnoritoku0424@gmail.com; 8Department of Surgery, National Hospital Organization Higashihiroshima Medical Center, Higashihiroshima 7390041, Japan; hr.hiroshimau@gmail.com; 9Department of Gastroenterological and Transplant Surgery, Graduate School of Biomedical and Health Sciences, Hiroshima University, Hiroshima 7398511, Japan; iop890@hiroshima-u.ac.jp (Y.S.); hohdan@hiroshima-u.ac.jp (H.O.); 10Department of Surgery, JA Hiroshima General Hospital, Hatsukaichi 7388503, Japan; sugiyama0113@gmail.com; 11Department of Surgery, Chuden Hospital, Hiroshima 7308562, Japan; mhikedaken@gmail.com; 12Department of Surgery, JR Hiroshima Hospital, Hiroshima 7320057, Japan; shishimasa@m3.dion.ne.jp; 13Department of Surgery, Chugoku Rosai Hospital, Kure 7370193, Japan; tsktfukuda@fch.ne.jp; 14Department of Surgery, Miyoshi Central Hospital, Miyoshi 7288502, Japan; kei.okano.tiko0404@hotmail.co.jp; 15Department of Surgery, Tsuchiya General Hospital, Hiroshima 7300811, Japan; ma_nishihara@tsuchiya-hp.jp

**Keywords:** gastric cancer, palliative surgery, stage IV, distal gastrectomy, gastrojejunostomy

## Abstract

**Simple Summary:**

For symptomatic stage IV gastric cancer involving major symptoms such as bleeding or obstruction, palliative surgery may be considered an option to relieve symptoms. Palliative gastrectomy or gastrojejunostomy is selected depending on the resectability of the primary tumor and/or surgical risk. However, treatment policies differ depending on the institution as to whether gastrectomy or gastrojejunostomy should be performed for symptomatic stage IV gastric cancer. We considered that gastrectomy might contribute more to prognosis than gastrojejunostomy for gastric cancer located in the middle or lower-third region where total gastrectomy can be avoided. Here, we compare the prognosis of gastrectomy and gastrojejunostomy for symptomatic stage IV gastric cancer. We demonstrate that distal gastrectomy for symptomatic stage IV gastric cancer located in the middle or lower-third regions contributes to prognosis with acceptable safety when compared to gastrojejunostomy.

**Abstract:**

Background: The prognostic prolongation effect of reduction surgery for asymptomatic stage IV gastric cancer (GC) is unfavorable; however, its prognostic effect for symptomatic stage IV GC remains unclear. We aimed to compare the prognosis of gastrectomy and gastrojejunostomy for symptomatic stage IV GC. Methods: This multicenter retrospective study analyzed record-based data of patients undergoing palliative surgery for symptomatic stage IV GC in the middle or lower-third regions between January 2015 and December 2019. Patients were divided into distal gastrectomy and gastrojejunostomy groups. We compared clinicopathological features and outcomes after propensity score matching (PSM). Results: Among the 126 patients studied, 46 and 80 underwent distal gastrectomy and gastrojejunostomy, respectively. There was no difference in postoperative complications between the groups. Regarding prognostic factors, surgical procedures and postoperative chemotherapy were significantly different in multivariate analysis. Each group was further subdivided into groups with and without postoperative chemotherapy. After PSM, the data of 21 well-matched patients with postoperative chemotherapy and 8 without postoperative chemotherapy were evaluated. Overall survival was significantly longer in the distal gastrectomy group (*p* = 0.007 [group with postoperative chemotherapy], *p* = 0.02 [group without postoperative chemotherapy]). Conclusions: Distal gastrectomy for symptomatic stage IV GC contributes to prognosis with acceptable safety compared to gastrojejunostomy.

## 1. Introduction

Gastric cancer (GC) is one of the most common malignancies and the third leading cause of cancer-related deaths worldwide. There were 782,685 GC-related deaths, accounting for approximately 8.2% of the total cancer deaths among 185 countries in 2018 [1]. In the past decade, median overall survival of approximately 12 months has been reported with chemotherapy alone [2,3,4,5]. Regarding reduction surgery for asymptomatic stage IV GC, in a systematic review by Mahar et al., the prognosis-improving effect of reduction surgery was not clearly observed [6]; however, many studies have reported that the prognosis-improving effect could be achieved with limited incurable factors [7,8,9]. However, in a subsequent prospective randomized controlled trial, there was no survival benefit of additional gastrectomy over chemotherapy alone (REGATTA) [10].

On the contrary, for symptomatic stage IV GC involving major symptoms such as bleeding or obstruction, palliative surgery may be considered an option to relieve symptoms. Palliative gastrectomy or gastrojejunostomy is selected depending on the resectability of the primary tumor and/or surgical risk [11]. However, treatment policies differ depending on the institution as to whether gastrectomy or gastrojejunostomy should be performed for symptomatic stage IV GC. We considered that gastrectomy might contribute more to prognosis than gastrojejunostomy for GC located in the middle or lower-third region where total gastrectomy can be avoided. A small number of retrospective studies have reported no survival benefits of gastrectomy compared to those of gastrojejunostomy for stage IV GC with gastric outlet obstruction, but the background factors were poorly matched [12,13]. Therefore, this retrospective study elucidated whether distal gastrectomy or gastrojejunostomy for symptomatic stage IV GC provides benefits to the patients by matching in terms of not only incurable factors but also inflammation and nutritional factors. In view of this, we aimed to determine the perioperative and oncological outcomes of distal gastrectomy as a palliative surgery for symptomatic stage IV GC and compared the data with those of gastrojejunostomy through propensity score matching analysis.

## 2. Materials and Methods

### 2.1. Study Design

We conducted a retrospective cohort study wherein we reviewed data from the medical records of patients with stage IV GC who underwent R2 surgery (distal gastrectomy or gastrojejunostomy) between January 2015 and December 2019 in 13 institutions belonging to the Hiroshima Surgical Study Group of Clinical Oncology (HiSCO), Hiroshima, Japan. We selected patients who met the following inclusion criteria: stage IV GC (excluding only positive abdominal lavage cytology as an incurable factor), symptoms (hemoglobin concentration <10 g/dL or obstruction), located in the middle or lower-third regions. Patients were excluded if they met any of the following criteria: pancreatic infiltration or severe duodenal development of GC, liver dysfunction (aspartate or alanine aminotransferase concentration >100 U/L or total bilirubin concentration >2 mg/dL), and moderate or higher quantities of ascites (exceeding the pelvic cavity, etc.). The Institutional Review Board of Onomichi General Hospital approved this study (OJH-202128).

### 2.2. Treatment and Procedure

Each physician decided the treatment procedure, such as surgical procedure (distal gastrectomy or gastrojejunostomy), indication of chemotherapy, chemotherapy regimen, and duration of chemotherapy.

### 2.3. Outcomes

We compared the perioperative and oncological outcomes between the gastrectomy and gastrojejunotomy groups with or without chemotherapy. The primary endpoint was overall survival. Operative time, bleeding, and duration of hospital stay after surgery were recorded. Surgical complications were evaluated according to the Clavien–Dindo (CD) classification.

### 2.4. Statistical Analyses

Continuous variables were presented as medians and ranges and compared between the groups using the Mann–Whitney U test. Categorical variables were presented as numbers and percentages and compared using Fisher’s exact test. Survival curves were generated using the Kaplan–Meier method and compared between different groups using the log-rank test. Multivariate analyses for survival were performed using Cox proportional hazards regression analysis. Variables with a *p*-value of <0.05 in univariate analysis were entered into multivariate analysis using Cox proportional hazards regression models. Hazard ratio (HR) and 95% confidence interval (CI) were used to estimate survival predictors. Differences between the results of comparative tests were considered statistically significant at two-sided *p* < 0.05.

To overcome bias due to the different distributions of covariates among patients from the distal gastrectomy groups and the gastrojejunal bypass groups with and without chemotherapy, propensity score matching analysis was performed using a multiple logistic regression model to predict the probability of each patient being allocated to a distal gastrectomy group based on clinicopathological variables.

To evaluate the discrimination and calibration abilities of the propensity scores, C statistics were used. The model showed good discrimination in the chemotherapy group (C statistic, 0.822 [95% CI, 0.728–0.915]; *p* < 0.01) and in the non-chemotherapy group (C statistic, 0.891 [95% CI 0.790–0.993]; *p* < 0.01).

A one-to-one matching algorithm without replacement was used, where all treated patients were matched to the closest control within a range of 0.20 standard deviations of the logit of the estimated propensity score. This matching was successful as the C statistic was well balanced (C statistic, 0.544 [95% CI 0.368–0.721]; *p* = 0.624, C statistic, 0.500 [95% CI 0.208–0.792]; *p* = 1.000, respectively). Data analyses were performed using SPSS software (version 27; IBM Corp., Armonk, NY, USA).

## 3. Results

Patients’ demographic and oncological characteristics and perioperative outcomes are shown in Table 1. Overall, 126 symptomatic patients who underwent palliative surgery for stage IV GC were included in this study; 46 patients had undergone distal gastrectomy, and the remaining 80 patients underwent gastrojejunostomy. Of the 126 patients, 76 received postoperative chemotherapy, and 50 did not receive postoperative chemotherapy. Although the operative time was shorter and blood loss was less in the gastrojejunostomy group, there were no significant differences in length of stay and postoperative complications between the groups. Regarding prognostic factors, American Society of Anesthesiologists Physical Status, neutrophil-to-lymphocyte ratio (NLR), prognostic nutritional index, modified Glasgow Prognosis Score, peritoneal metastasis, number of metastasis factors, postoperative chemotherapy, surgical approach, surgical procedure, operative blood loss, and length of hospital stay were significant factors in univariate analysis. In multivariate analysis, only postoperative chemotherapy and surgical procedures were significant prognostic factors (Table 2). In all cases, distal gastrectomy had a significantly better prognosis than gastrojejunostomy (*p* ≤ 0.001) (Figure 1a).

Of the 76 patients who received postoperative chemotherapy, 31 underwent distal gastrectomy, and 45 underwent gastrojejunostomy. Performance Status (PS) was better and preoperative obstruction, NLR, distant lymph node metastasis, and use of laparoscopic approach were higher in the gastrojejunostomy group (Table 3). After propensity score matching with PS, obstruction, NLR, distant lymph node metastasis and surgical approach for 76 patients who received postoperative chemotherapy, distal gastrectomy and gastrojejunostomy matched 21 cases each (Table 4). The overall survival of the two groups before and after propensity score matching is shown in Figure 1b. After matching, the median survival time was 13.3 months in the gastrojejunostomy group and 22.0 months in the distal gastrectomy group. The 24-month survival rate was 4.8% in the gastrojejunostomy group and 49.7% in the distal gastrectomy group (HR 0.406, *p* = 0.008).

Of the 50 patients who did not receive postoperative chemotherapy, 15 underwent distal gastrectomy, and 35 underwent gastrojejunostomy. Preoperative obstruction was higher, and NLR was higher in the gastrojejunostomy group (Table 3). After propensity score matching with obstruction and NLR for 50 patients who did not receive postoperative chemotherapy, distal gastrectomy and gastrojejunostomy required 8 matched cases for each category (Table 4). The overall survival of the two groups before and after propensity score matching is shown in Figure 1c. After matching, the median survival time was 2.6 months in the gastrojejunostomy group and 7.0 months in the distal gastrectomy group. None of the patients survived for 24 months in either group, but the prognosis was significantly prolonged in the distal gastrectomy group (HR 0.289, *p* = 0.026).

A similar study was conducted in 98 cases with gastric outlet obstruction (Appendix A). In multivariate analysis, only postoperative chemotherapy, surgical procedures and length of hospital stays were significant prognostic factors (Appendix A). In all cases with gastric outlet obstruction, distal gastrectomy had a significantly better prognosis than gastrojejunostomy (*p* < 0.001) (Appendix A).

Of the 58 patients who received postoperative chemotherapy, 18 underwent distal gastrectomy, and 40 underwent gastrojejunostomy. Distant lymph node metastasis was higher in the gastrojejunostomy group (Appendix A). After propensity score matching with distant lymph node metastasis for 58 patients who received postoperative chemotherapy, distal gastrectomy and gastrojejunostomy matched 13 cases each (Appendix A). The overall survival of the two groups before and after propensity score matching is shown in Appendix A. After matching, distal gastrectomy had a significantly better prognosis than gastrojejunostomy (*p* = 0.007) (Appendix A).

Of the 40 patients who did not receive postoperative chemotherapy, 7 underwent distal gastrectomy, and 33 underwent gastrojejunostomy. NLR was higher in the gastrojejunostomy group (Appendix A). After propensity score matching with NLR for 40 patients who did not receive postoperative chemotherapy, distal gastrectomy and gastrojejunostomy matched 6 cases each (Appendix A). The overall survival of the two groups before and after propensity score matching is shown in Appendix A. After matching, distal gastrectomy had a significantly better prognosis than gastrojejunostomy (*p* = 0.003) (Appendix A).

## 4. Discussion

In this study, distal gastrectomy significantly prolonged overall survival compared to gastrojejunostomy in patients with symptomatic stage IV GC located in the middle or lower-third region. This result was the same with or without postoperative chemotherapy. Similar results were obtained by examining only cases with gastric outlet obstruction. There are two possible explanations for the prognosis-prolonging effect of gastrectomy. First, tumor volume reduction may have extended the prognosis, as can be inferred from the prognosis-prolonging effect obtained even in cases without chemotherapy. Second, it is suggested that chemotherapy compliance may be improved by excising the primary gastric tumor in symptomatic patients. Although not significant, gastrectomy has been reported to provide a higher rate of solid intake than gastrojejunostomy [14], which may lead to improved chemotherapy compliance. In addition, in the subgroup analysis of the REGATTA study, compliance with chemotherapy was maintained in GC located in the lower-third region, which avoided total gastrectomy, resulting in comparable overall survival [10]. Conversely, palliative total gastrectomy should be performed with caution, as it can reduce chemotherapy compliance and can worsen prognosis.

Regarding perioperative outcomes, distal gastrectomy showed significantly greater surgical time and bleeding volume than gastrojejunostomy. On the contrary, there was no significant difference in the length of hospital stay after surgery or the occurrence of complications of CD3 or higher. It can be said that distal gastrectomy can be safely performed even in patients with stage IV GC.

Similar studies conducted in the past were confounded by selection bias because patients with good PS, fewer comorbidities, better nutritional status, less inflammation, and smaller tumor burden were more likely to undergo gastrectomy [12,13]. In our study, we compared gastrectomy and gastrojejunostomy by matching nutritional and inflammatory conditions, as well as PS and tumor factors, and then proved the survival benefits of gastrectomy. This is the first study to compare distal gastrectomy with gastrojejunostomy by matching not only incurable factors but also inflammatory and nutritional factors in multiple institutions over a relatively short period.

In addition to obstruction, this study also included cases of anemia (hemoglobin concentration of <10 g/dL). It is also an important clinical question as to whether gastrectomy or gastrojejunostomy with incomplete transection should be performed in cases with tumor bleeding. Because more than half of the patients in this study were anemic, palliative gastrectomy of anemic patients was considered to have greater benefits to patients than gastrojejunostomy.

To improve oral intake for gastric outlet obstruction in GC, gastrointestinal stent placement is also a candidate, along with distal gastrectomy and gastrojejunostomy. There is no consensus on the overall survival in gastrointestinal stent placement and gastrojejunostomy [15,16]. Keranen et al. reported that palliative gastrectomy seems to provide a survival benefit in contrast to gastrointestinal stent placement and gastrojejunostomy to treat gastric outlet obstruction. Palliative resection should be considered a treatment option for patients suitable for surgery [17]. Our study suggests that gastrectomy may improve prognosis over gastrojejunostomy if the condition is tolerable to general anesthesia surgery, but there is no evidence to compare the long-term prognosis of distal gastrectomy with gastrointestinal stents. Gastrointestinal stenting, gastrojejunostomy, and distal gastrectomy are all options for improving oral intake in gastric outlet obstruction, and in clinical practice, they are selected according to the case background.

There are some limitations to our study. First, this was a retrospective study, and the number of cases after PSM was not large. Second, background matching of tumor factors may be inadequate. Stage IV GC has various oncological conditions. The degree of liver metastasis, lymph node metastasis, and peritoneal dissemination also varied. In this study, cases of massive ascites and liver dysfunction were excluded, and an attempt was made to indirectly match the oncological background with the nutrition and inflammation scores. However, this alone may not be sufficient for oncological background matching. It may be beneficial to use the stage IV GC classification proposed by Yoshida et al. [18,19] In the future, prospective studies are needed to confirm these results.

## 5. Conclusions

In our retrospective study, distal gastrectomy for symptomatic stage IV GC contributes to better prognosis with acceptable safety compared to gastrojejunostomy. However, it is difficult to completely align the background of stage IV GC, and randomized controlled trials are warranted to fill a gap.

## Figures and Tables

**Figure 1 cancers-14-00388-f001:**
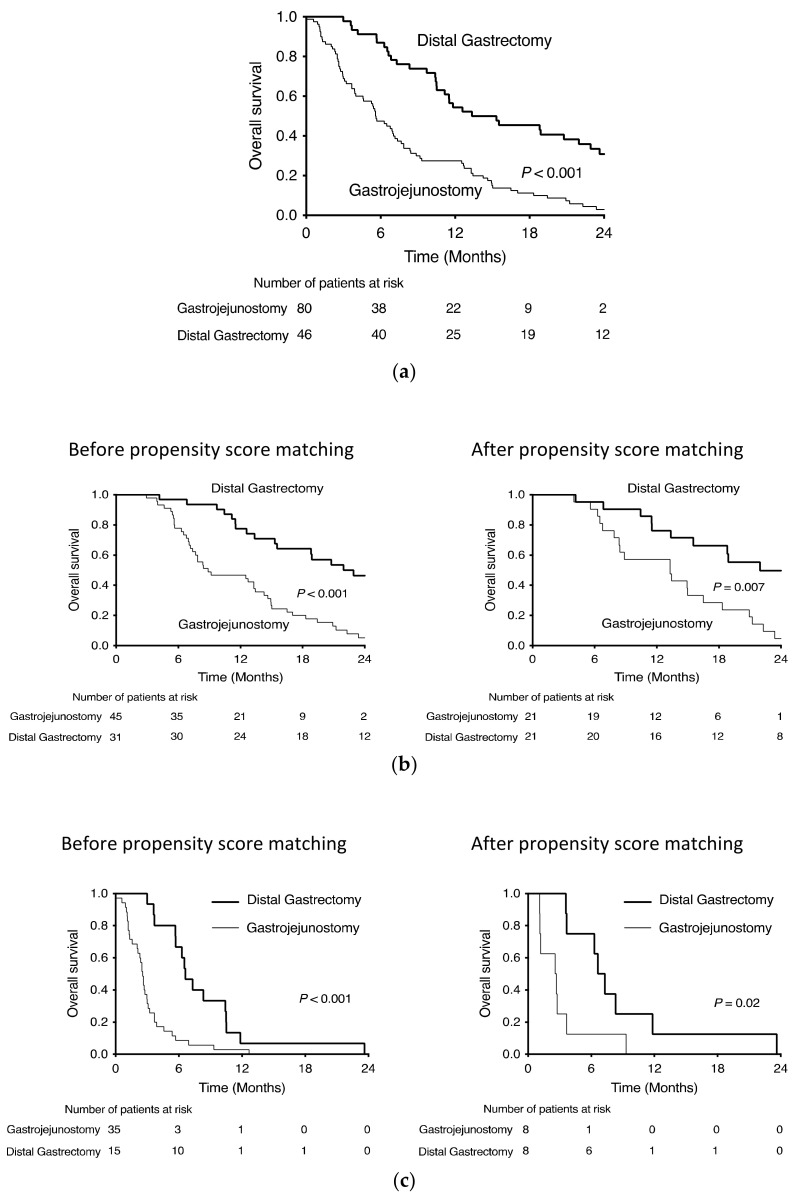
Kaplan–Meier survival curves of distal gastrectomy and gastrofejunostomy for stage IV gastric cancer (**a**) all cases, (**b**) with chemotherapy and (**c**) without chemotherapy.

**Table 1 cancers-14-00388-t001:** General characteristics of 126 GC patients.

Variables	Heading	Distal Gastrectomy	Gastrojejunostomy	*p*-Value
		(*n* = 46)	(*n* = 80)	
Age	<70	19 (41.3%)	28 (35.0%)	0.567
	≥70	27 (58.7%)	52 (65.0%)	
Sex	Male	31 (67.4%)	55 (68.8%)	1.000
	Female	15 (32.6%)	25 (31.3%)	
BMI	<25	41 (89.1%)	70 (87.5%)	1.000
	≥25	5 (10.9%)	10 (12.5%)	
ASA PS	1	0 (0%)	2 (2.5%)	0.544
	2	34 (73.9%)	53 (66.3%)	
	3	12 (26.1%)	25 (31.3%)	
PS	0	27 (58.7%)	46 (57.5%)	0.178
	1	16 (34.8%)	18 (22.5%)	
	2	2 (4.3%)	11 (13.8%)	
	3	1 (2.2%)	5 (6.3%)	
Anemia	Present	33 (71.7%)	51 (63.7%)	0.434
	Absent	13 (28.3%)	29 (36.3%)	
Obstruction	Present	25 (54.3%)	73 (91.3%)	**<0.001**
	Absent	21 (45.7%)	7 (8.8%)	
CEA	<5	27 (60.0%)	38 (48.1%)	0.262
	≥5	18 (40.0%)	41 (51.9%)	
	unknown	1	1	
CA19-9	<37	26 (59.1%)	45 (57.0%)	0.851
	≥37	18 (40.9%)	34 (43.0%)	
	unknown	2	1	
NLR	<3	28 (60.9%)	22 (27.5%)	**<0.001**
	≥3	18 (39.1%)	58 (72.5%)	
PNI	<40	21 (45.7%)	48 (60.0%)	0.139
	≥40	25 (54.3%)	32 (40.0%)	
mGPS	0	13 (31.7%)	15 (19.5%)	0.174
	1–2	28 (68.3%)	62 (80.5%)	
	unknown	5	3	
Macroscopic type	Non-4	42 (91.3%)	66 (82.5%)	0.198
	4	4 (8.7%)	14 (17.5%)	
Histologic type	Intestinal	23 (50.0%)	30 (37.5%)	0.193
	Diffuse	23 (50.0%)	50 (62.5%)	
Invasion of adjacent organs	Present	1 (2.2)	9 (11.3%)	0.092
	Absent	45 (97.8%)	71 (88.8%)	
Distant lymph node metastasis	Present	10 (21.7%)	28 (35.0%)	0.158
	Absent	36 (78.3%)	52 (65.0%)	
Liver metastasis	Present	14 (30.4%)	21 (26.3%)	0.681
	Absent	32 (69.6%)	59 (73.8%)	
Peritoneal metastasis	Present	27 (58.7%)	59 (73.8%)	0.111
	Absent	19 (41.3%)	21 (26.3%)	
Number of metastasis factors	1	38 (82.6%)	48 (60.0%)	0.071
	2	6 (13.0%)	23 (28.7%)	
	3	1 (2.2%)	6 (7.5%)	
	4	1 (2.2%)	3 (3.8%)	
Postoperative Chemotherapy	Present	31 (67.4%)	45 (56.3%)	0.259
	Absent	15 (32.6%)	35 (43.8%)	
Surgical approach	Open	41 (89.1%)	63 (78.7%)	0.154
	Laparoscopic	5 (10.9%)	17 (21.3%)	
Operative time	(min)	233 (118–366)	126 (61–268)	**<0.001**
Blood loss	(ml)	132.5 (6–680)	12.5 (0–940)	**<0.001**
Hospital stays	(days)	13.5 (8–138)	17.5 (1–72)	0.445
Complications ≥CD3	Present	4 (8.7%)	7 (8.8%)	1.000
	Absent	42 (91.3%)	73 (91.3%)	

Variables in bold are statistically significant (*p* < 0.05). BMI, body mass index; ASA PS, American Society of Anesthesiologists physical status; PS, Performance Status; CEA, carcinoembryonic antigen; CA19-9, carbohydrate antigen 19-9; NLR, neutrophil-to-lymphocyte ratio; PNI, prognostic nutritional index; mGPS, modified Glasgow Prognostic Score; CD, Clavien-Dindo Classification.

**Table 2 cancers-14-00388-t002:** Univariate and multivariate analysis of overall survival.

Variables	Heading	Univariate	*p*-Value	Multivariate	*p*-Value
		HR (95%CI)		HR (95%CI)	
Age	<70	1			
	≥70	1.348 (0.926–1.961)	0.119		
Sex	Male	1			
	Female	1.189 (0.807–1.752)	0.381		
BMI	<25	1			
	≥25	0.909 (0.518–1.593)	0.738		
ASA PS	1–2	1		1	
	3	1.600 (1.076–2.378)	**0.020**	1.362 (0.870–2.133)	0.177
PS	0	1			
	1–3	1.320 (0.922–1.918)	0.127		
Anemia	Present	0.928 (0.632–1.363)	0.704		
	Absent	1			
Obstruction	Present	1.566 (0.999–2.455)	0.050		
	Absent	1			
CEA	<5	1			
	≥5	1.366 (0.945–1.974)	0.097		
CA19–9	<37	1			
	≥37	1.199 (0.823–1.748)	0.344		
NLR	<3	1		1	
	≥3	1.942 (1.316–2.867)	**<0.001**	1.388 (0.824–2.339)	0.218
PNI	<40	1		1	
	≥40	0.567 (0.389–0.826)	**0.003**	0.732 (0.436–1.228)	0.237
mGPS	0	1		1	
	1–2	1.861 (1.168–2.964)	**0.009**	1.272 (0.717–2.259)	0.411
Macroscopic type	Non-4	1			
	4	1.113 (0.673–1.841)	0.677		
Histologic type	Intestinal	1			
	Diffuse	1.341 (0.912–1.915)	0.141		
Invasion of adjacent organs	Present	1.053 (0.548–2.023)	0.876		
	Absent	1			
Distant lymph node metastasis	Present	1.231 (0.829–1.828)	0.302		
	Absent	1			
Liver metastasis	Present	0.978 (0.649–1.473)	0.915		
	Absent	1			
Peritoneal metastasis	Present	1.512 (1.026–2.250)	**0.041**	1.034 (0.647–1.653)	0.889
	Absent	1		1	
Number of metastasis factors	1	1		1	
	2–4	1.519 (1.031–2.239)	**0.035**	1.043 (0.638–1.704)	0.867
Postoperative Chemotherapy	Present	0.180 (0.119–0.274)	**<0.001**	0.172 (0.104–0.284)	<0.001
	Absent	1		1	
Surgical approach	Open	1		1	
	Laparoscopic	1.908 (1.189–3.062)	**0.007**	1.011 (0.562–1.818)	0.971
Surgical procedure	Distal Gastrectomy	0.379 (0.255–0.565)	**<0.001**	0.263 (0.146–0.475)	**<0.001**
	Gastrojejunostomy	1		1	
Operative time	<152	1			
	≥152	0.738 (0.513–1.062)	0.102		
Blood loss	<30	1		1	
	≥30	0.682 (0.474–0.982)	**0.040**	1.447 (0.857–2.444)	0.167
Hospital stays	<16	1		1	
	≥16	1.944 (1.334–2.833)	**<0.001**	1.320 (0.810–2.150)	0.266
Complications ≥CD3	Present	1.653 (0.886–3.087)	0.114		
	Absent	1			

HR, hazard ratio; CI, confidential index. Variables in bold are statistically significant (*p* < 0.05).

**Table 3 cancers-14-00388-t003:** General characteristics before propensity score matching.

		With Chemotherapy			Without Chemotherapy		
Variables	Heading	Distal Gastrectomy	Gastrojejunostomy	*p*-Value	Distal Gastrectomy	Gastrojejunostomy	*p*-Value
		(*n* = 31)	(*n* = 45)		(*n* = 15)	(*n* = 35)	
Age	<70	17 (54.8%)	18 (40.0%)	0.245	2 (13.3%)	10 (28.6%)	0.304
	≥70	14 (45.2%)	27 (60.0%)		13 (86.7%)	25 (71.4%)	
Sex	Male	23 (74.2%)	33 (73.3%)	1.000	8 (53.3%)	22 (62.9%)	0.547
	Female	8 (25.8%)	12 (26.7%)		7 (46.7%)	13 (37.1%)	
BMI	<25	28 (90.3%)	38 (84.4%)	0.514	13 (86.7%)	32 (91.4%)	0.629
	≥25	3 (9.7%)	7 (15.6%)		2 (13.3%)	3 (8.6%)	
ASA PS	1	0 (0%)	1 (2.2%)	0.557	0 (0%)	1 (2.9%)	0.833
	2	26 (83.9%)	33 (73.3%)		8 (53.3%)	20 (57.1%)	
	3	5 (16.1%)	11 (24.4%)		7 (46.7%)	14 (40.0%)	
PS	0	18 (58.1%)	36 (80.0%)	**0.037**	9 (60.0%)	10 (28.6%)	0.220
	1	12 (38.7%)	6 (13.3%)		4 (26.7%)	12 (34.3%)	
	2	1 (3.2%)	3 (6.7%)		1 (6.7%)	8 (22.9%)	
	3	0 (0%)	0 (0%)		1 (6.7%)	5 (14.3%)	
Anemia	Present	22 (71.0%)	25 (44.4%)	0.231	11 (73.3%)	26 (74.3%)	1.000
	Absent	9 (29.0%)	20 (55.6%)		4 (26.7%)	9 (25.7%)	
Obstruction	Present	18 (58.1%)	40 (88.9%)	**0.003**	7 (46.7%)	33 (94.3%)	**<0.001**
	Absent	13 (41.9%)	5 (11.1%)		8 (53.3%)	2 (5.7%)	
CEA	<5	20 (64.5%)	22 (50.0%)	0.244	7 (50.0%)	16 (45.7%)	1.000
	≥5	11 (35.5%)	22 (50.0%)		7 (50.0%)	19 (54.3%)	
	unknown	0	1		1	0	
CA19-9	<37	19 (63.3%)	24 (54.5%)	0.482	7 (50.0%)	21 (60.0%)	0.542
	≥37	11 (36.7%)	20 (45.5%)		7 (50.0%)	14 (40.0%	
	unknown	1	1		1	0	
NLR	<3	19 (61.3%)	16 (35.6%)	**0.036**	9 (60.0%)	6 (17.1%)	**0.006**
	≥3	12 (38.7%)	29 (64.4%)		6 (40.0%)	29 (82.9%)	
PNI	<40	12 (38.7%)	25 (55.6%)	0.168	9 (60.0%)	23 (65.7%)	0.754
	≥40	19 (61.3%)	20 (44.4%)		6 (40.0%)	12 (34.3%)	
mGPS	0	10 (38.5%)	8 (19.0%)	0.095	3 (20.0%)	7 (20.0%)	1.000
	1–2	16 (61.5%)	34 (81.0%)		12 (80.0%)	28 (80.0%)	
	unknown	5	3		0	0	
Macroscopic type	Non-4	29 (93.5%)	37 (82.2%)	0.185	13 (86.7%)	29 (82.9%)	1.000
	4	2 (6.5%)	8 (17.8%)		2 (13.3%)	6 (17.1%)	
Histologic type	Intestinal	16 (51.6%)	16 (35.6%)	0.237	7 (46.7%)	14 (40.0%)	0.759
	Diffuse	15 (48.4%)	29 (64.4%)		8 (53.3%)	21 (60.0%)	
Invasion of adjacent organs	Present	1 (3.2%)	5 (11.1%)	0.391	0 (0%)	4 (11.4%)	0.302
	Absent	30 (96.8%)	40 (88.9%)		15 (100%)	31 (88.6%)	
Distant lymph node metastasis	Present	5 (16.1%)	19 (42.2%)	**0.023**	5 (33.3%)	9 (25.7%)	0.733
	Absent	26 (83.9%)	26 (57.8%)		10 (66.7%)	26 (74.3%)	
Liver metastasis	Present	10 (32.3%)	12 (26.7%)	0.616	4 (26.7%)	9 (25.7%)	1.000
	Absent	21 (67.7%)	33 (73.3%)		11 (73.3%)	26 (74.3%)	
Peritoneal metastasis	Present	18 (58.1%)	30 (66.7%)	0.477	9 (60.0%)	29 (82.9%)	0.146
	Absent	13 (41.9%)	15 (33.3%)		6 (40.0%)	6 (17.1%)	
Number of metastasis factors	1	26 (83.9%)	27 (60.0%)	0.056	12 (80.0%)	21 (60.0%)	0.203
	2	5 (16.1%)	12 (26.7%)		1 (6.7%)	11 (31.4%)	
	3	0 (0%)	5 (11.1%)		1 (6.7%)	1 (2.9%)	
	4	0 (0%)	1 (2.2%)		1 (6.7%)	2 (5.7%)	
Surgical approach	Open	31 (100%)	37 (82.2%)	**0.018**	10 (66.7%)	26 (74.3%)	0.733
	Laparoscopic	0 (0%)	8 (17.8%)		5 (33.3%)	9 (25.7%)	

Variables in bold are statistically significant (*p* < 0.05).

**Table 4 cancers-14-00388-t004:** General characteristics after propensity score matching.

		With Chemotherapy			Without Chemotherapy		
Variables	Heading	Distal Gastrectomy	Gastrojejunostomy	*p*-Value	Distal Gastrectomy	Gastrojejunostomy	*p*-Value
		(*n* = 21)	(*n* = 21)		(*n* = 8)	(*n* = 8)	
Age	<70	11 (52.4%)	8 (38.1%)	0.536	1 (12.5%)	2 (25.0%)	1.000
	≥70	10 (47.6%)	13 (61.9%)		7 (87.5%)	6 (75.0%)	
Sex	Male	17 (81.0%)	14 (66.7%)	0.484	3 (37.5%)	5 (62.5%)	0.619
	Female	4 (19.0%)	7 (33.3%)		5 (62.5%)	3 (37.5%)	
BMI	<25	18 (85.7%)	16 (76.2%)	0.697	7 (87.5%)	7 (87.5%)	1.000
	≥25	3 (14.2%)	5 (23.8%)		1 (12.5%)	1 (12.5%)	
ASA PS	1	0 (0%)	1 (4.8%)	0.454	0 (0%)	1 (12.5%)	1.000
	2	18 (85.7%)	15 (71.4%)		4 (50.0%)	3 (37.5%)	
	3	3 (14.3%)	5 (23.8%)		4 (50.0%)	4 (50.0%)	
PS	0	13 (61.9%)	16 (76.2%)	0.734	3 (37.5%)	1 (12.5%)	0.804
	1	7 (33.3%)	4 (19.0%)		3 (37.5%)	4 (50.0%)	
	2	1 (4.8%)	1 (4.8%)		1 (12.5%)	1 (12.5%)	
	3	0 (0%)	0 (0%)		1 (12.5%)	2 (25.0%)	
Anemia	Present	12 (57.1%)	14 (66.7%)	0.751	4 (50.0%)	2 (25.0%)	0.608
	Absent	9 (42.9%)	7 (33.3%)		4 (50.0%)	6 (75.0%)	
Obstruction	Present	17 (81.0%)	16 (76.2%)	1.000	6 (75.0%)	6 (75.0%)	1.000
	Absent	4 (19.0%)	5 (23.8%)		2 (25.0%)	2 (25.0%)	
CEA	<5	16 (76.2%)	9 (45.0%)	0.058	4 (50.0%)	3 (37.5%)	1.000
	≥5	5 (23.8%)	11 (55.0%)		4 (50.0%)	5 (62.5%)	
	unknown	0	1		0	0	
CA19-9	<37	13 (65.0%)	13 (65.0%)	1.000	4 (50.0%)	5 (62.5%)	1.000
	≥37	7 (35.0%)	7 (35.0%)		4 (50.0%)	3 (37.5%)	
	unknown	1	1		0	0	
NLR	<3	12 (57.1%)	11 (52.4%)	1.000	6 (75.0%)	6 (75.0%)	1.000
	≥3	9 (42.9%)	10 (47.6%)		2 (25.0%)	2 (25.0%)	
PNI	<40	9 (42.9%)	10 (47.6%)	1.000	4 (50.0%)	4 (50.0%)	1.000
	≥40	12 (57.1%)	11 (52.4%)		4 (50.0%)	4 (50.0%)	
mGPS	0	5 (29.4%)	4 (21.1%)	0.706	2 (25.0%)	1 (12.5%)	1.000
	1–2	12 (70.6%)	15 (78.9%)		6 (75.0%)	7 (87.5%)	
	unknown	4	2		0	0	
Macroscopic type	Non-4	19 (90.5%)	17 (81.0%)	0.663	6 (75.0%)	6 (75.0%)	1.000
	4	2 (9.5%)	4 (19.0%)		2 (25.0%)	2 (25.0%)	
Histologic type	Intestinal	7 (33.3%)	7 (33.3%)	1.000	2 (25.0%)	1 (12.5%)	1.000
	Diffuse	14 (66.7%)	14 (66.7%)		6 (75.0%)	7 (87.5%)	
Invasion of adjacent organs	Present	1 (4.8%)	5 (23.8%)	0.184	0 (0%)	2 (25.0%)	0.467
	Absent	20 (95.2%)	16 (76.2%)		8 (100%)	6 (75.0%)	
Distant lymph node metastasis	Present	5 (23.8%)	3 (14.3%)	0.697	2 (25.0%)	2 (25.0%)	1.000
	Absent	16 (76.2%)	18 (85.7%)		6 (75.0%)	6 (75.0%)	
Liver metastasis	Present	5 (23.8%)	6 (28.6%)	1.000	2 (25.0%)	2 (25.0%)	1.000
	Absent	16 (76.2%)	15 (71.4%)		6 (75.0%)	6 (75.0%)	
Peritoneal metastasis	Present	12 (57.1%)	16 (76.2%)	0.326	5 (62.5%)	6 (75.0%)	1.000
	Absent	9 (42.9%)	5 (23.8%)		3 (37.5%)	2 (25.0%)	
Number of metastasis factors	1	17 (81.0%)	15 (71.4%)	0.608	6 (75.0%)	4 (50.0%)	0.413
	2	4 (19.0%)	4 (19.0%)		1 (12.5%)	3 (37.5%)	
	3	0 (0%)	2 (9.5%)		1 (12.5%)	0 (0%)	
	4	0 (0%)	0 (0%)		0 (0%)	1 (12.5%)	
Surgical approach	Open	21 (100%)	21 (100%)	-	5 (62.5%)	7 (87.5%)	0.569
	Laparoscopic	0 (0%)	0 (0%)		3 (37.5%)	1 (12.5%)	

## Data Availability

The data presented in this study are available on request from the corresponding author. The data is not publicly available due to patient privacy and the General Data Protection Regulation.

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
