# Peer review of "Distal Gastrectomy for Symptomatic Stage IV Gastric Cancer Contributes to Prognosis with Acceptable Safety Compared to Gastrojejunostomy"

_cancers, 2022, doi:10.3390/cancers14020388_

Round 1

Reviewer 1 Report

I, the reviewer believe that this revision has increased the value of your article. Unfortunately, since it is only a retrospective study, the potential for selection bias by the surgeon cannot be denied, and the scientific credibility of the article is not necessarily high, but the content is very suggestive and should be considered for publication.

Reviewer 2 Report

The reviewer has red the revised manuscript. 

I have now understood that the authors considered my comments when they revised their manuscript. I would like to make the decission to "Accept".

Reviewer 3 Report

I agree with the authors' comments and found the manuscript to be suitable for publication.

This manuscript is a resubmission of an earlier submission. The following is a list of the peer review reports and author responses from that submission.

Round 1

Reviewer 1 Report

This is a very interesting study, but unfortunately, there is a fundamental problem that makes it difficult to accept it as it is.

It is well known that radical gastrectomy without preoperative treatment for stage IV gastric cancer does not improve the prognosis. However, palliative surgery for symptomatic patients is beneficial in reducing subjective symptoms. Basically, the surgical procedure is palliative gastrectomy if it can be resectable, and bypass surgery if it is unresectable or if the patient wants to avoid invasion. Therefore, there are differences in the target patients between the two procedures that cannot be measured by PS, NLR, mGPS, and PNI. Although patients with direct pancreatic invasion were excluded in this study, it seems that patients who selected gastrojejunostomy had more cases of pyloric stenosis, and those who had gastrectomy had more bleeding. It is well known that the prognosis of patients with pyloric stenosis is poor. Therefore, even if PSM was performed, the comparison of this study was not fair.

If a fair comparison is to be made, the two surgical procedures should be compared only in cases where the occupying site is the lower third of the stomach and palliative surgery is decided due to the symptom of pyloric stenosis. Stage IV cases other than pyloric stenosis should not be included.

Reviewer 2 Report

The authors address an topic that has important clinical implications for oncologists and surgeons. Despite the obvious study limitations due to the small sample size, I found that the overall message is sound and I agree with the fact that partial gastrectomy is superior to gastrojejunostomy in selected patients. The manuscript is well written and balanced. 

Reviewer 3 Report

The reviewer really appreciates having an opportunity to evaluate this manuscript entitled “Distal gastrectomy for symptomatic stage IV gastric cancer 3 contributes to prognosis with acceptable safety compared to 4 gastrojejunostomy” written by Fujikuni et al of HiSCO. First of all, the manuscript is well written and is easy for readers to understand.

It is a big problem whether patients with symptomatic stage IV gastric cancer should undergo gastrectomy. REGATTA trial teaches surgeons that palliative gastrectomy does not provide patients with metastatic gastric cancer a better survival outcome. One of the interpretations regarding the important results of this study is that patients undergone gastrectomy did not undergo postoperative chemotherapy sufficiently. Thus, if patients who underwent sufficient chemotherapy after gastrectomy had been randomly assigned, the results might have been different, that is, patients who received chemotherapy after surgery might have better survival outcomes. Gastrectomy for patients with symptomatic stage IV gastric cancer is one of the solutions to relieve their symptoms and may prolong their survival.

As the authors mentioned in Discussion section, survival outcomes may be influenced by many factors, such as patient general condition, nutritional status, inflammatory status, metastatic status. In this study, the authors matched backgrounds of patients using propensity scores. Thus, the results of this study are more reliable than those of previous studies. However, the reviewer is still anxious about potential heavy biases in this study. Basically, surgeons remove tumor if the tumor is easy to be removed or does not have so many metastases unless surgeons decide that they never remove stage IV tumors even if the tumor is easy to be removed when patients have some symptoms. Although the authors matched some patient factors and tumor factors, the degree of metastasis, such as the number of liver metastasis, the grade of peritoneal metastasis (so called P1, P2 and P3) and the extent of adjacent organ invasion were not matched. Furthermore, some patients in gastrectomy group may undergo R0 resection. For instance, surgeons performed gastrectomy with hepatectomy for patients with 1 to 3 liver metastases, but they never performed it for patients with huge or numerous liver metastasis, which is equally described as presence of liver metastasis in this study. The survival outcomes of patients who underwent gastrectomy are very favorable, while those are relatively worse compared with current results of far advanced gastric cancer patients who received chemotherapy. The reviewer would say that these considerations and results strongly indicate that surgeons naturally selected patients whose tumors were easy to be removed and whose survivals seemed relatively favorable. The reviewer understands that they are according to the retrospective nature and that the authors well understand the issue, of course. However, the reviewer is very sorry to say that the reviewer disagrees with the conclusions based the design, method and results of this study.